# Biosynthetic Mesh Reconstruction after Abdominoperineal Resection for Low Rectal Cancer: Cross Relation of Surgical Healing and Oncological Outcomes: A Multicentric Observational Study

**DOI:** 10.3390/cancers15102725

**Published:** 2023-05-11

**Authors:** Claudio Gambardella, Federico Maria Mongardini, Menelaos Karpathiotakis, Francesco Saverio Lucido, Francesco Pizza, Salvatore Tolone, Simona Parisi, Giusiana Nesta, Luigi Brusciano, Antonio Gambardella, Ludovico Docimo, Massimo Mongardini

**Affiliations:** 1Division of General, Oncological, Mini-Invasive and Obesity Surgery, University of Study of Campania “Luigi Vanvitelli”, 80131 Naples, Italy; f.mongardini@gmail.com (F.M.M.); francescosaverio.lucido@unicampania.it (F.S.L.); salvatore.tolone@unicampania.it (S.T.); simona.parisi@unicampania.it (S.P.); giusiana.nesta@gmail.com (G.N.); luigi.brusciano@unicampania.it (L.B.); ludovicodocimo75@gmail.com (L.D.); 2Division of General Surgery, Policlinico Umberto I, Sapienza University of Rome, 00185 Rome, Italy; menelaokadras@gmail.com (M.K.); massimo.mongardini@uniroma1.it (M.M.); 3Department of Surgery, Hospital ‘A. Rizzoli’, Lacco Ameno, 80076 Naples, Italy; francesco.pizza@gmail.com; 4Department of Precision Medicine, University of Campania “L. Vanvitelli”, 80138 Naples, Italy; antonio.gambardella@unicampania.it

**Keywords:** abdominoperineal resection, rectal cancer, biosynthetic meshes, perineal wound reconstruction

## Abstract

**Simple Summary:**

The large perineal defect, with impaired wound healing and delayed start of the adjuvant chemotherapy, can make the reconstructive phase of abdominoperineal resection for low rectal cancer extremely challenging. Using biosynthetic mesh for the neo-perineum reconstruction after a Miles’ procedure is a poorly investigated technique, which, in our series, resulted in safe, reproducible results affected by limited complications. Moreover, for improved perineal wound healing, it guaranteed a faster start of the adjuvant therapy with clear reduction in oncological outcomes (i.e., recurrences and death).

**Abstract:**

Background: Local wound complications are among the most relevant sequelae after an abdominoperineal resection (APR) for low rectal cancer. One of the proposed techniques to improve the postoperative recovery and to accelerate the initiation of adjuvant chemotherapy is the mesh reinforcement of the perineal wound. The aim of the current study is to compare the surgical and oncological outcomes after APR performed with a biosynthetic mesh reconstruction versus the conventional procedure. Methods: From 2015 to 2020, in two tertiary centres, the surgical outcomes, the wound events (i.e., surgical site infections, wound dehiscence and the complete healing time) and the oncological outcomes (i.e., time length to start adjuvant chemo-radiotherapy, an over 8-week delay in chemotherapy and the recurrence rate) were retrospectively analysed in patients undergoing APR reinforced with biosynthetic mesh (Group A) and conventional APR (Group B). Results Sixty-one patients were treated with APR (25 in Group A and 36 in Group B). Patients in Group A presented lower time for: healing (16 versus 24 days, *p* = 0.015), inferior perineal wound dehiscence rates (one versus nine cases, *p* = 0.033), an earlier adjuvant therapy start (26 versus 70 days, *p* = 0.003) and a lower recurrence rate (16.6% vs. 33.3%, *p* = 0.152). Conclusions: In our series, the use of a biosynthetic mesh for the neo-perineum reconstruction after a Miles’ procedure has resulted in safe, reproducible results affected by limited complications, guarantying a rapid start of the adjuvant therapy with clear benefits in oncological outcomes. Further randomized clinical trials with long-term follow-up are needed to validate these results.

## 1. Introduction

Malignant disease of the colon and rectum is one of the most frequent cancers in Western countries, and despite the advances of oncological therapies, the surgical approach remains a milestone for its treatment [1,2]. In recent decades, the scientific community has continuously provided relevant evolutions in surgical techniques with a widespread adoption of minimally invasive approaches, such as laparoscopic and robotic resections, transanal endoscopic microsurgery (TEM) and transanal total mesorectal excision (TATME) for low rectal cancers [3,4]. Nevertheless, in 15–20% of the cases, due to the localization of the disease (within 5 cm of the anal verge), the infiltration of the anal sphincter muscles or the impossibility to achieve a negative resection margin (5 cm proximally and 2 cm distally), a conservative technique preserving the perineal plane and the sphincteric structures is precluded [5].

These cases require a more demolitive surgery, such as abdominoperineal resection (APR or Miles’ procedure), consisting in a permanent colostomy after the removal of the sigmoid colon, rectum and anus. Despite using the laparoscopic or robotic approach, the large perineal defect can make the reconstructive phase of this procedure extremely challenging, especially when an extralevator abdominoperineal excision (ELAPE) is needed; the process involves dissecting outside the extralevator muscles plane and removing the entire pelvic floor to achieve oncological radicality [6]. Therefore, perineal wound closure and healing is often impaired, potentially with the onset of serious complications, leading to a delay of the adjuvant oncological therapy; therapy that is proven to be crucial in reducing the distal recurrence rate and increasing the patient’s disease-free survival [7,8,9,10,11]. Moreover, the risk of perineal wound complications after APR could be even higher after neo-adjuvant chemoradiotherapy (CHT-RT), which is often performed in distal rectal cancer patients. The surgical complications may include intra-abdominal or pelvic abscesses, hematoma, wound dehiscence, perineal hernia and delayed wound healing [12,13,14]. Evidence-based medicine and international guidelines suggest starting the adjuvant chemotherapy within 6–8 weeks after the successful completion of surgery, once complete or near-complete healing of the perineal wound has been achieved [15].

The main surgical techniques for perineal reconstruction after APR, in addition to the direct closure of the perineal wound where it is technically possible, include fascial-myocutaneous flaps, omentoplasty, tissue expanders, silicone breast implants and synthetic or biological prostheses [16,17,18,19,20].

Materials used in synthetic prostheses, such as polypropylene, create visceral adhesions in contact with the peritoneum and the pelvic floor organs that can cause erosions, fistulas, or perforations; for this reason, these meshes are not recommended in perineal wound reconstruction. On the other hand, biological prostheses, despite being more advantageous, have limitations related to low resistance and reduced integration into the tissue (ingrowth), the tendency of shrinking, high costs, the risk of rejection reactions or transmission of viral infections and the need of tissue tracking and preoperative soaking [21,22,23].

The use of biosynthetic prosthetic materials in recent years has revolutionized the pelvic–perineal reconstructive surgery, achieving satisfactory results. These prostheses have a chemical structure very similar to the natural polymers contained in biological tissues, such as collagen, which gives these materials excellent biocompatibility. The interaction between the three-dimensional matrix of the synthetic copolymers and human tissue promotes cell adhesion and cell growth. In some circumstances, the body “metabolizes” the polymer by degrading it over time until it is completely reabsorbed after tissue regeneration. This is the case of the GORE^®^ BIO-A^®^ Tissue Reinforcement [W. L. Gore & Associates, Inc., Newark, DE, USA] prosthetic material, which was applied in our study.

The aim of the current observational retrospective study is to analyse and compare the surgical and oncological outcomes in patients undergoing Miles’ APR and perineal wound reconstruction with or without the use of an absorbable biosynthetic prosthesis in two high-volume tertiary centres.

## 2. Materials and Methods

### 2.1. Study Design

This study is reported according to the STROBE statement for cohort studies [24] and was conducted according to the ethical principles stated in the Declaration of Helsinki. Written informed consent was obtained from all patients.

### 2.2. Study Setting and Study Population

From 1 January 2015 to 31 December 2020, all the patients undergoing Miles’ APR at the Division of General Surgery of the Policlinico Umberto I of Rome and the Division of General, Oncologic, Mininvasive and Obesity Surgery of Vanvitelli University of Naples, were considered in the study. Inclusion criteria were presence of low rectal cancer (within 5 cm of the anal verge) or anal cancer with local invasion of the sphincter muscles complex and the perineal body and with an American Society of Anesthesiologists (ASA) physical status of grade ≤ III [25]. Exclusion criteria were inflammatory bowel disease, neurological diseases, ongoing corticosteroid therapies and previous episodes of prosthetic material rejection.

All patients received a routine preoperative clinical and instrumental diagnostic assessment with anamnestic data collection, accurate general and proctological examination, a colonoscopy, laboratory blood tests and cardiological and anaesthetic evaluations. The oncological staging comprised a total body computed tomography scan with contrast medium and a pelvic floor MRI. Each case was discussed, prior to surgery, at a weekly multidisciplinary meeting of the colorectal unit, composed of oncologists, surgeons, radiologists, radiotherapists and gastroenterologists.

After the referral for surgery, each patient received a detailed explanation of the procedure by the medical staff and had to sign a personalised informed consent form. All the operations were performed by the same experienced surgeons.

Clinical data were collected in an electronic database and retrospectively analysed. Patients who underwent the APR with the biosynthetic mesh reconstruction of the perineal wound were considered in Group A, while patients undergoing the conventional APR procedure with direct closure of the perineal wound were considered in Group B. In Group A, a synthetic biocompatible and resorbable mesh (GORE BIO-A Tissue Reinforcement, 9 × 15 cm) was applied to replace and reinforce the perineum, restoring the anatomy and the containment function of the pelvic floor (Figure 1). This prosthesis consists of a highly porous 3D matrix that serves as a scaffold for tissue regeneration and is composed of a copolymer (polyglycolic acid: trimethylene carbonate [PGA:TMC]) that is gradually absorbed by the body. After implantation, the bioabsorbable material undergoes hydrolytic degradation over a period of about 6 months, leaving behind no synthetic material, and the PGA:TMC also encourages the development of vascularized tissue. As a result, the mesh is replaced by a layer of regenerated tissue that reinforces the perineal wound closure [26,27,28,29].

The postoperative follow-up period included regular surgical control in outpatient clinics until wound healing and an oncological assessment every 6 months for the first 2 years.

### 2.3. Surgical Technique

#### 2.3.1. Conventional Abdominoperineal Resection

Antibiotic prophylaxis with ceftriaxone sodium salt and metronidazole was administered to all patients intra- and postoperatively. All the surgical procedures were performed under general anaesthesia and in a double surgical equipe: one for the abdominal laparoscopic approach and one for the perineal dissection. Therefore, the dissection of the pelvic rectum proceeded in both directions simultaneously, with the two operators performing a kind of rendezvous at the level of the elevator plane, with considerable advantages for the patient [18,19,20].

After a complete exploration of the abdomen, an APR with standard Miles’ technique was performed. The abdominal approach concluded with complete mobilization of the pelvic rectum and a total excision of the mesorectum (TME).

Once the dissection of the rectum was completed beyond the levator plane, the perineal surgeon coordinated with the abdominal team for the dissection of the perineal rectum. After the specimen was removed, a terminal colostomy was positioned in the left iliac fossa and the pelvic drains were placed. In Group B we adopted a direct closure of the neo-perineum, once we verified that the perineal tissues were not subjected to tension and did not distort the local anatomy, especially in female patients. The residues of the elevator complex of the anus were firstly approximated with absorbable sutures; then, the subcutaneous and ischiorectal fat, and finally the skin, were sutured in distinct layers, while trying to avoid the formation of unfilled cavities and re-establish the architecture of the pelvic floor. A suction drain was positioned.

#### 2.3.2. Abdominoperineal Resection with Mesh Reinforcement of the Perineal Plane

In Group A, the surgical procedure was the same except for the perineal plane reconstruction, which was achieved by the interposition of the biosynthetic mesh GORE^®^ BIO-A^®^ Tissue Reinforcement. The prosthesis was placed laparoscopically, close to the plane of the residual levator muscles, and fixed with detached resorbable stitches or fibrin glue along the pelvic peritoneal free margin, the residual Toldt’s fascia and the presacral fascia at its most distal margin (Figure 2 and Figure 3). Therefore, the closure of the perineal wound was completed by suturing the musculocutaneous flaps and the skin directly. The dimensions of the mesh were 9 × 15 cm in diameter in all cases, ideal to create a scaffold to replace the damaged pelvic fascia and support the pelvic organs.

### 2.4. Outcome Measures

Mean operative time was reported in minutes. Perineal wound events were classified as surgical-site infections according to Centers for Disease Control and Prevention (CDC) criteria (superficial, deep or organ space) [30,31]. Interventions for wound events were categorized as antibiotic treatment, bedside wound interventions, percutaneous drainage or surgical debridement. Sero-mucous discharge was evaluated every week within 2 months of surgery and was expressed in centilitres (cc). Postoperative pain was evaluated at 1, 3, 6 and 12 months after surgery using a visual analogic scale (VAS). Time needed to complete perineal wound healing was expressed in days.

The presence of any other postoperative complications (i.e., occlusion, haemorrhage, parastomal hernia or perineal hernia) was assessed at the outpatient clinics. Particularly, perineal hernia was defined as any clinically visible or palpable protuberance in the perineal region. In that case, the diagnosis was confirmed by a computed tomography (CT) scan.

Regarding the oncological outcomes, the time length to start the adjuvant chemo-radiotherapy was recorded and expressed in days, the delay of over 8 weeks to chemotherapy was expressed in number of cases in each group and the recurrence, evaluated each 6 months by a CT scan, was expressed in rates.

### 2.5. Study Outcomes

The primary aim of the study was the evaluation of the postoperative surgical outcomes, especially the time needed for wound healing, the presence of wound events and the postoperative complications in the patients of the two groups. The secondary aim was the assessment of oncological outcomes (i.e., the time length to start the adjuvant chemo-radiotherapy, numbers of patients who delayed the chemotherapy for over 8 weeks, the recurrence rate, and the mortality) in the patients of both groups.

### 2.6. Statistical Analysis

Statistical analysis was performed via Excel 2011^®^ (Microsoft, Redmond, WA, USA) and through the Graph-Pad Prism^®^ 9 program (San Diego, CA, USA). Categorical data were reported as raw numbers with percentages in parenthesis. Continuous data were reported as means ± standard deviation or as medians with the range in parenthesis, according to the distribution. The differences between results were analysed by the unpaired *t*-test if they were summarized as means, the Mann–Whitney U test if they were summarized as medians or the Fisher’s exact test if they were reported as percentages. A *p* value of less than 0.05 was considered statistically significant.

## 3. Results

From 1 January 2015 to 31 December 2020, of the 92 patients referred for low rectal cancer, 61 met the inclusion criteria and received an APR as the primary surgical treatment. Twenty-five patients underwent APR with biosynthetic mesh reinforcement (Group A) while thirty-six had a conventional APR procedure (Group B) (Figure 4).

All cases were diagnosed with a primary cancer. Neo-adjuvant CHT-RT was reported for 48 of 61 patients (19 in Group A and 29 in Group B). The patients with persistent rectal bleeding, abscessed neoplasms, acquired immunodeficiency and those refusing preoperative treatment were not eligible for neo-adjuvant CHT-RT.

Of the 61 patients included in the study, 34 were males (55.7%) and 27 were females (44.3%), with a median age of 68.3 years (36–85 years) and body mass index (BMI) of 28.85 kg/m^2^ (22–38 kg/m^2^). Demographic and pathological findings are reported in Table 1.

Mean operative time and intraoperative blood loss are detailed in Table 2. The median hospitalization was 6 days (1–3 days) in Group A and 8 days (6–18 days) in Group B (*p* = 0.572). No relevant intraoperative complications occurred in Group A, while a case of intraoperative bleeding requiring conversion to laparotomy was verified in Group B. The death of one female patient was registered in Group A on the third postoperative day due to deep vein thrombosis and massive pulmonary embolism. No perioperative mortality was reported in the patients of Group B (Table 3).

### 3.1. Primary Outcome

Postoperative wound events are illustrated in Table 4. Hematoma and seroma were conservatively managed with percutaneous drainage in four and three cases in Group A and B, respectively. Superficial incisional infection occurred in two cases in Group A (8.3%) and eight cases in Group B (22.2%), while deep incisional infection occurred in only three patients in Group B (8.3%); all of them were treated with surgical bed-side debridement and daily irrigation and dressing of the wound, along with intravenous antibiotic therapy.

Partial dehiscence of the perineal wound and the formation of a presacral collection/abscess marked the postoperative course in one male patient in Group A, while this occurred in nine patients in Group B (*p* = 0.033). All were conservatively managed at outpatient clinics with the use of advanced dressings. In five cases in Group B, a vacuum assisted closure (VAC) therapy was applied, achieving the closure of the wound within 60 days. The median time needed for complete wound healing was lower in Group A (16 days vs. 24 days) with a statistically significant difference (*p* = 0.015).

No cases of prosthesis removal or rejection were reported in Group A.

Median postoperative pain evaluated with the VAS score was significantly lower (*p* < 0.05) in Group A at 1,3,6 and 12 months of follow-up after surgery (Figure 5). Two cases of perineal hernia occurred in Group B (5.5%, *p* = 0.240) and were confirmed by CT scan; nevertheless, the patients denied further surgical treatment.

### 3.2. Secondary Outcome

The median time before undergoing adjuvant chemotherapy was significantly lower in Group A (26 days vs. 70 days, *p* = 0.003). Moreover, four and sixteen patients in Group A and B, respectively, reported an over 8-week delay in the initiation of the adjuvant chemotherapy (*p* = 0.025). At a median follow-up period of 30–31 months, four patients (16.6 %) in Group A presented a recurrence, while in Group B that occurred in twelve patients (33.3%) (*p* = 0.152); however, no statistically relevant differences in recurrence and mortality rates between the two groups were reported (Table 5).

## 4. Discussion

Perineal wound healing after Miles’ procedure for low rectal cancer still represents a great challenge for colorectal surgeons. The extensive demolition of the pelvic floor often results in a loss of the structure and function of the connected anatomical districts.

Primary closure of large defects or secondary intention healing are often burdened with major complications [32,33,34,35,36], such as excessive scar tension on surrounding tissues, intracavitary hematoma and seroma and herniation of abdominal organs. Thus, the possibility of a pelvic floor reconstruction after APR is highly recommended to restore the anatomical and functional integrity of the pelvic floor and preserve the function of the involved organs, especially for the female lower genital tract [37,38,39].

Different surgical techniques have been proposed for post-Miles’ perineal reconstruction but there is no consensus among the experts due to the lack of data from prospective randomized trials and long-term follow-up. In their multicentric study, Musters G.D. et al. (2014) [40] reported their experience with post-APR omentoplasty, concluding that it has the role of replenishing rather than the function to guarantee reinforcement of a reconstructed pelvic floor. Jensen K.K. et al. (2014) found that the use of fascio-myocutaneous flaps is associated with an unacceptably high risk of perineal hernias after APR [41].

The use of prosthetic materials in pelvic–perineal reconstructive surgery has become increasingly prominent in recent years, inspired by their consolidated use in repairing abdominal hernias and emphasising the importance of reinforcing the deteriorated tissues with external support. However, Cui et al. (2009) claimed that non-resorbable, synthetic expanded polytetrafluoroethylene (PTFE) prosthesis might have good tissue tolerance and compliance but demonstrates a poor resistance to infection [42]. The same authors, furthermore, concluded that the re-intervention rate for perineal wound complications could potentially be halved by using resorbable biological prostheses (from 816% to 5–10%) and, consequently, the incidence of perineal hernias could be reduced to 10% [43].

Various biosynthetic meshes have been recently introduced, representing a cost-effective option compared to biological materials. In this category, GORE^®^ BIO-A^®^ Tissue Reinforcement represents a co-polymer prosthesis (polyglycolic acid and trimethylene carbonate [PGA:TMC]), that has been widely adopted in different surgical fields with excellent results [26,27].

To our knowledge, this is the first study in the literature analysing the surgical and oncological outcomes of patients undergoing APR with a biosynthetic perineal mesh reconstruction. In our series, the Miles’ procedure with mesh reinforcement (Group A) resulted effective in reducing the incidence of postoperative wound events. The median time needed for complete wound healing was significantly lower in Group A (16 days versus 24 days, *p* = 0.015). Despite superficial and deep wound infection rates being higher in patients undergoing conventional APR, there was no statistically significant difference. Overall, the local complications were negligible in patients with biosynthetic mesh reinforcement of the neo-perineum and no cases of rejection or removal of the prosthesis were verified. In addition, the median postoperative pain evaluated with VAS scores was significantly lower in Group A patients. These results confirm the role of biosynthetic mesh reinforcement in improving and accelerating the wound healing process in patients who undergo an APR.

Moreover, the patients of Group A presented a lower incidence of perineal wound dehiscence (one versus nine patients, *p* = 0.033). This should be considered a relevant success obtained by the use of the prosthetic material that, in the reported series, allowed us to address the patients to oncological therapies earlier, limiting one of the most frequent and bothersome sequelae after APR.

Patients with mesh reinforcement after APR (Group A), due to the accelerated perineal wound healing, had the possibility to start the adjuvant oncological treatment earlier, within 3–4 weeks after surgery (26 versus 70 days, *p* = 0.003). Regarding the long-term outcomes, the recurrence and the mortality rates were also lower in Group A (16.6% versus 33.3% and 8.3% versus 11.1%, respectively), but without showing a statistical significance in both cases. Noteworthily, the small number of cases limited the statistical power for identifying the differences between long-term oncological outcomes.

As reported in the literature, for patients with over stage II rectal cancer, adjuvant chemotherapy is recommended [9] as it improves the survival rate. The survival outcomes, in fact, are reported to be increasingly lower if the adjuvant therapy begins over 12 weeks after surgery [31]. The strict association of survival in colorectal cancer with the time of starting adjuvant treatment was underlined by Biagi et al. (2011), as they claimed that the overall survival rate decreases by 14% for every 4-week delay to the initiation of adjuvant chemotherapy [15]. Therefore, the perineal wound complications occurring after APR negatively affect the possibility to start a timely adjuvant protocol and, consequently, decrease the survival rate.

Furthermore, not all reconstructive techniques after APR are able to accelerate wound healing and the start of the adjuvant chemotherapy. As stated by Althumari et al. (2016), the perineal reconstruction with flaps did not reduce the length of time to initiate chemotherapy; therefore, this procedure did not influence the oncological outcomes [43].

In our series, the APR performed with biosynthetic mesh reinforcement has shown to be a safe, effective and easily reproducible procedure with irrelevant complications compared to conventional surgery. This technique contributed not only to a functional reconstruction of the neo-perineum, but also permitted a significantly faster wound healing process with a reduced rate of dehiscence and a shorter time length to start the adjuvant therapy. Despite the promising results, there are some limitations in our study: the retrospective design, the limited number of patients enrolled and the follow-up period of less than 5 years.

## 5. Conclusions

Perineal wound healing and dehiscence are the main concerns after APR for rectal cancer, as they can significantly delay the start of adjuvant chemotherapy and, thus, negatively affect the recurrence and long-term survival rates. Biosynthetic mesh reinforcement after Miles’ procedure could be one of the most promising techniques to overcome this issue. In our series, this procedure allowed rapid wound healing associated with inferior dehiscence rates, reducing, therefore, the costs of complications management. Furthermore, the quality of life of the patients was improved by anticipating the initiation of the adjuvant treatment and reducing the time needed for perineal wound healing. Nonetheless, further, larger-cohort randomized trials with longer follow-up periods are necessary to validate these results.

## Figures and Tables

**Figure 1 cancers-15-02725-f001:**
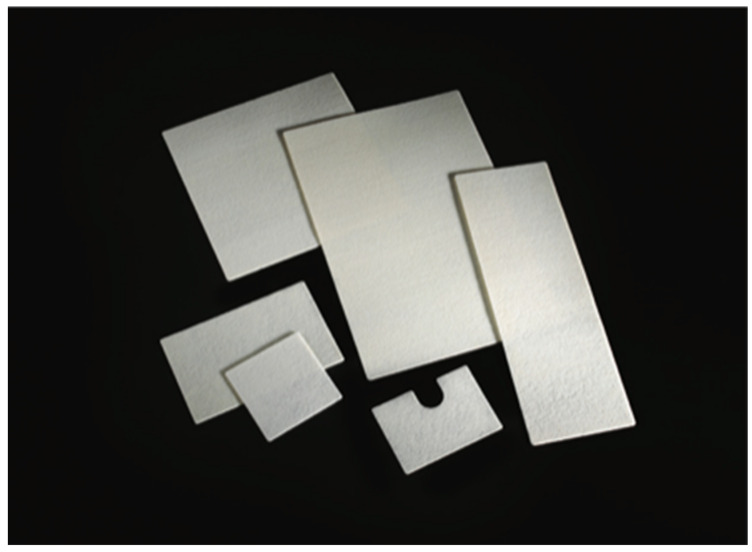
GORE^®^ BIO-A^®^ Tissue Reinforcement meshes.

**Figure 2 cancers-15-02725-f002:**
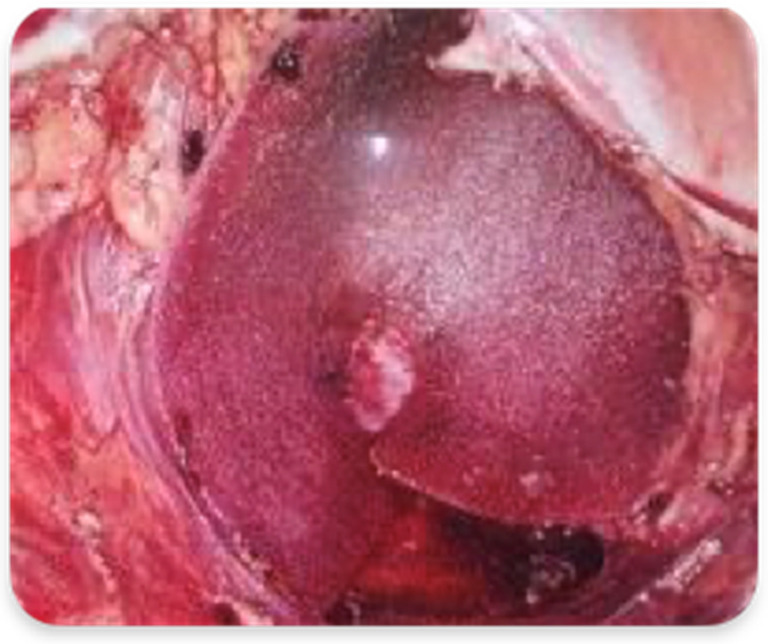
Mesh GORE^®^ BIO-A^®^ Tissue Reinforcement fixed with fibrin glue.

**Figure 3 cancers-15-02725-f003:**
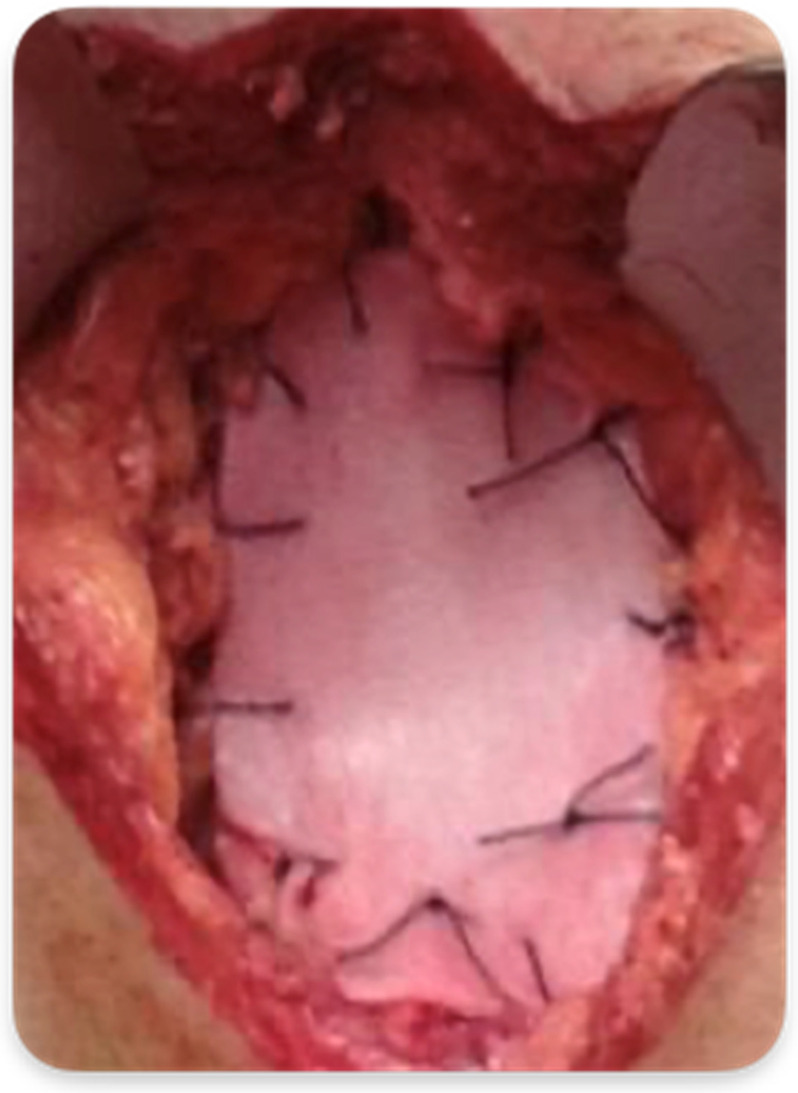
Mesh GORE^®^ BIO-A^®^ Tissue Reinforcement fixed with resorbable stiches.

**Figure 4 cancers-15-02725-f004:**
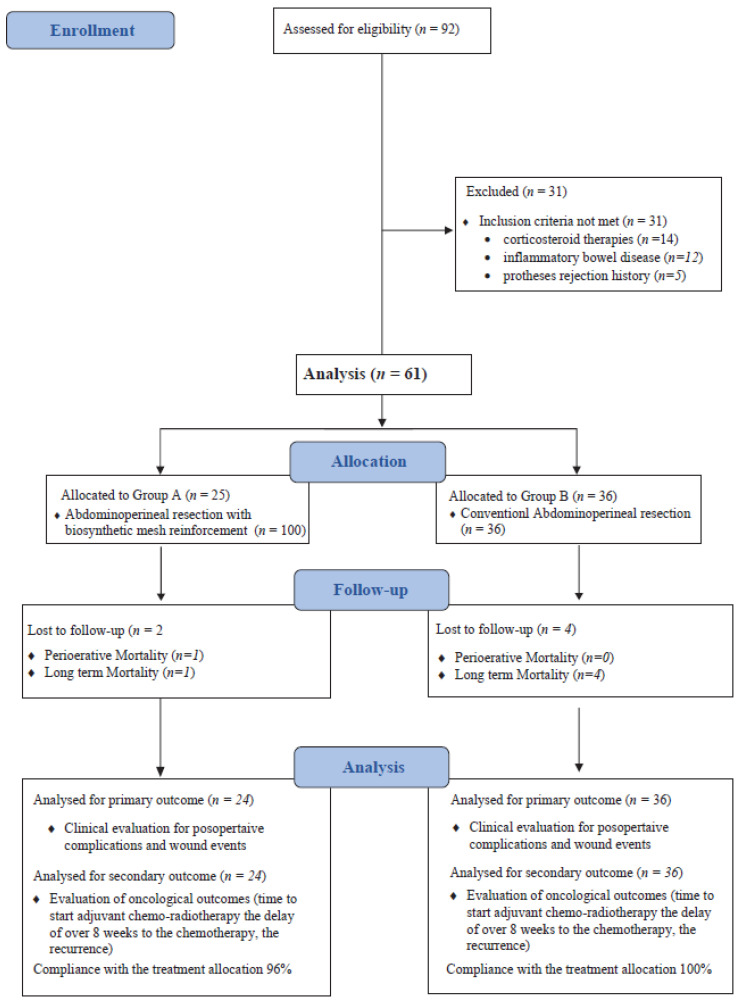
CONSORT diagram.

**Figure 5 cancers-15-02725-f005:**
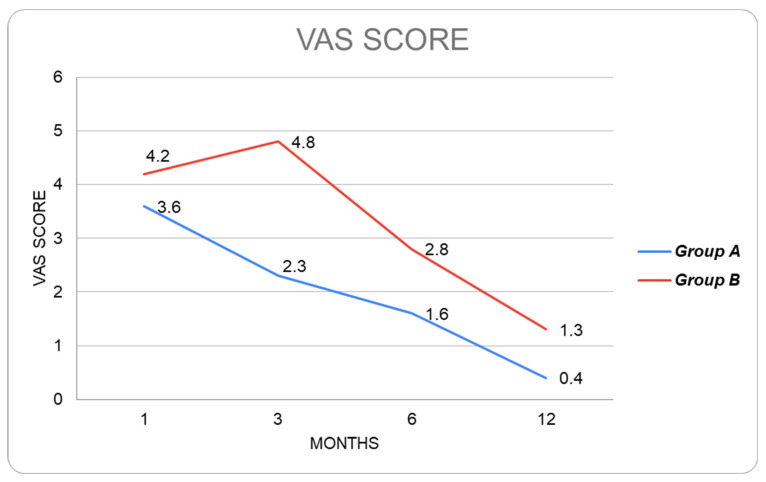
Postoperative pain evaluation using VAS scores in Group A and Group B.

**Table 1 cancers-15-02725-t001:** Patients’ demographics and epidemiological characteristics.

	Group A 25 Patients	Group B 36 Patients	*p*
Gender (Male/Female)	13/12 (52%/48%)	21/15 (58.3%/41.7%)	0.624
Age (Years) °	69.3 (36–85)	67.6 (43–78)	0.160
BMI	28.2 (22–35.7)	29.3 (24–38)	0.304
ASA I-II/III-IV	18/7 (72%/28%)	19/17 (52.7%/47.3%)	0.130
Hypertension	16 (64%)	26 (72.2%)	0.495
Diabetes	9 (36%)	14 (38.9%)	0.818
Chronic obstructive pulmonary disease	4 (16%)	7 19.4%)	0.730
Cerebrovascular Disease	0	1 (2.7%)	0.391
Smoking	9 (36%)	15 (41.6%)	0.655
Chronic kidney disease	1 (4%)	2 (5.5%)	0.800
Heart ischemic attack	2 (8%)	2 (5.5%)	0.704
Preoperative symptoms			
Rectorrhagia Tenesmus Constipation Cachexia	21 (84%) 11 (44%) 18 (72%) 3 (12%)	29 (80.5%) 15 (41.6%) 28 (77%) 5 (13.8%)	0.730 0.856 0.606 0.829
Tumour Stage			
Stage I Stage II Stage III Stage IV	0 7 (28%) 16 (64%) 2 (8%)	0 14 (38.9%) 18 (50%) 4 (11.1%)	- 0.378 0.279 0.688
Tumour dimension (Centimetres)	6.3 (3–9)	5 (2–7)	0.347
Height from the anal margin (Centimetres)	2.2 (0–3)	2.5 (0–3)	0.734
Neo-adjuvant CHT-RT	19 (76%)	29 (80.5%)	0.512

° Values are expressed as number of cases or medians and a range. CHT-RT (Chemoradiotherapy).

**Table 2 cancers-15-02725-t002:** Perioperative outcomes in Group A and B.

	Group A 25 Patients	Group B 36 Patients	*p*
Operative time (min) °	235 (160–330)	195 (140–295)	0.125
Intraoperative blood loss (mL) °	275 (140–730)	230 (110–650)	0.423
Intraoperative complications	0	1 (Bleeding)	0.391
Hospitalization (days) °	6 (4–12)	8 (6–18)	0.572
Perioperative mortality	1 (4%)	0	0.226

° Values are expressed as number of cases or medians and a range.

**Table 3 cancers-15-02725-t003:** Morbidity (Clavien–Dindo classification) and mortality of patients of Group A and B.

	Until Discharge		Until 30 Days		After 30 Days	
	Group A	Group B	Group A	Group B	Group A	Group B
Uneventful postoperative course	>95%	>99%	>85%	>60%	>95%	>95%
Readmission	-	-	<1%	<1%	<1%	<1%
Reoperation	<1%	<1%	<1%	<1%	<1%	<1%
Clavien–Dindo Grade I-II	<1%	<1%	<15%	<40%	<1%	<1%
Clavien–Dindo Grade >III	<4%	<1%	<1%	<1%	<90%	<85%
Mortality	1	0	0	0	2	4
Bowel leak	0	0	0	0	0	0
Small bowel obstruction/internal hernia	0	0	0	0	0	0
Bleeding	0	0	0	0	0	0
Wound infection	0	0	2	11	0	0
Perineal wound dehiscence	0	0	1	9	0	0

**Table 4 cancers-15-02725-t004:** Postoperative wound events and surgical site infections in the two groups.

	Group A 24 Patients	Group B 36 Patients	*p*
Hematoma	4 (16.6%)	5 (13.8%)	0.767
Seroma	7 (29.1%)	6 (16.6%)	0.147
Superficial incisional infections	2 (8.3%)	8 (22.2%)	0.157
Deep incisional infections	0	3 (8.3%)	0.146
Perineal wound dehiscence	1 (4.2%)	9 (25%)	0.033 *
Time needed for wound healing (Days) °	16 (12–26)	24 (18–32)	0.015 *

° Values are expressed as number of cases or medians and a range. * Statistical Significant

**Table 5 cancers-15-02725-t005:** Oncological outcomes in Group A and B.

	Group A 24 Patients	Group B 36 Patients	*p*
Time to start adjuvant chemotherapy (days) °	26 (15–40)	70 (30–120)	0.003 *
Delay in chemotherapy (more than 8 weeks)	4 (16.6%)	16 (44.4%)	0.025 *
Mortality	2 (8.3%)	4 (11.1%)	0.725
Long-term recurrence	4 (16.6%)	12 (33.3%)	0.152
Follow-up (months) °	30 (28–32)	31 (27–33)	0.645

° Values are expressed as number of cases or medians and a range.* Statistical Significant

## Data Availability

The datasets used and/or analysed during the current study are available from the corresponding author on reasonable request.

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
