# Peer review of "Biosynthetic Mesh Reconstruction after Abdominoperineal Resection for Low Rectal Cancer: Cross Relation of Surgical Healing and Oncological Outcomes: A Multicentric Observational Study"

_cancers, 2023, doi:10.3390/cancers15102725_

Round 1

Reviewer 1 Report

This is a retrospective study evaluating the surgical and oncological outcomes of patients with low rectal cancer undergoing APR with or without biosynthetic mesh reconstruction. The manuscript is very clear, concise, and well written. The methods are good, and the results are well presented.

However, I have several minor questions/critiques for the authors to respond before considering the manuscript for final publication.

Introduction

·         - I would suggest the Authors to revise the sentence “which is often mandatory in distal rectal cancer patients” when discussing on neoadjuvant chemoradiotherapy. The term “indicated” would be better than mandatory. Recent studies have proved that we might be overusing neoadjuvant chemoradiotherapy, precision surgery alone can be sufficient in a good portion of patients.

·       -  Typo “such us” change to “such as”

·        - When showing an industry product as the mesh you need to put in brackets (Company name, city, country). Please correct while speaking o Gore BioA

·         -Typo “underwent Miles” change to “undergoing Miles”

Methods

·         -Typo “underwent Miles” change to “undergoing Miles”

·         - Statistical analysis: the authors show the data as median (range) or median (interquartile range)?

Results

·         - A consort figure showing the inclusion/exclusion criteria with the group division would benefit the study.

·         - The authors state “mean age of 68.3…” is it mean or median? Please check and revise

·         - Typo “bleeding required” change to “bleeding requiring”

·         - The authors are showing oncological results, therefore it is necessary to show the pathological staging (TNM, tumor dimension, height from anal verge, circumferential and distal resection margin). Please add this information also in the tables.

·        - Authors should also show the complications classified with the Clavien Dindo grading.

·         - The authors state that VAS was significantly different between groups. What was the p value?

·         - “The median time before undergo adjuvant chemotherapy” correct to “The median time before undergoing adjuvant chemotherapy”

Table 1

·         - Check for typos

·         - Please use the same terminology as for the text. CHT-RT not NeoAdjuvant CH-R

The English language requires minor check.

Author Response

This is a retrospective study evaluating the surgical and oncological outcomes of patients with low rectal cancer undergoing APR with or without biosynthetic mesh reconstruction. The manuscript is very clear, concise, and well written. The methods are good, and the results are well presented.

Thank you for your kind comments

However, I have several minor questions/critiques for the authors to respond before considering the manuscript for final publication.

Introduction

  • - I would suggest the Authors to revise the sentence “which is often mandatory in distal rectal cancer patients” when discussing on neoadjuvant chemoradiotherapy. The term “indicated” would be better than mandatory. Recent studies have proved that we might be overusing neoadjuvant chemoradiotherapy, precision surgery alone can be sufficient in a good portion of patients.
  • - Typo “such us” change to “such as”
  • - When showing an industry product as the mesh you need to put in brackets (Company name, city, country). Please correct while speaking o Gore BioA
  • -Typo “underwent Miles” change to “undergoing Miles”

Thank you for your precious suggestions. All the typos and grammatical errors were modified accordingly.

Methods

  • -Typo “underwent Miles” change to “undergoing Miles”

Thank you again for your suggestions. All the typos and grammatical errors were modified accordingly.

  • - Statistical analysis: the authors show the data as median (range) or median (interquartile range)?

Thank you for the opportunity to clarify. It was adopted the median and range as detailed in the statistical analyisis and in the tables.

Results

  • - A consort figure showing the inclusion/exclusion criteria with the group division would benefit the study.

Thank you for the suggestions. The consort figure was added accordingly.

  • - The authors state “mean age of 68.3…” is it mean or median? Please check and revise

It was revised.

  • - Typo “bleeding required” change to “bleeding requiring”

Thank you it was modified accordingly.

  • - The authors are showing oncological results, therefore it is necessary to show the pathological staging (TNM, tumor dimension, height from anal verge, circumferential and distal resection margin). Please add this information also in the tables.

Thank you for your preciuos suggestion. This data were added in table 1.

  • - Authors should also show the complications classified with the Clavien Dindo grading.

The Clavien Dindo table was added accordingly.

  • - The authors state that VAS was significantly different between groups. What was the p value?

Thabk you for the opportunity to clarify. The p Value was added in the results section.

  • - “The median time before undergo adjuvant chemotherapy” correct to “The median time before undergoing adjuvant chemotherapy”

It was modified accordingly.

Table 1

  • - Check for typos
  • - Please use the same terminology as for the text. CHT-RT not NeoAdjuvant CH-R

The tables were modified accordingly. The term CHT-RT was adopted in the text to indicate only the chemoradiotherapy. In this case we referred to neoadjuvant chemoradiotherapy. This was clarified in tables and in the text.

Comments on the Quality of English Language

The English language requires minor check.

An English profreading was performed by a native speaker.

Reviewer 2 Report

Congratulations for your interesting manuscript.

I'd suggest some minor points. Could you provide the information about presentation of the disease in each group (primary or recurrent)?

Regarding the results, in my point of view, the most relevant finding is the significant difference between the rates of wound dehiscence. This finding could be highlighted in the abstract, where is neither cited, and in the discussion and conclusions.

On the other hand, the difference in the recurrence rate, although twice in the group B, was not significant. I think that the study has no power to identify statistically differences on long term oncological outcomes. So I would just described this findings, but not highlighted as a significant result.

Author Response

Congratulations for your interesting manuscript.

Thank you for your kind comment.

I'd suggest some minor points. Could you provide the information about presentation of the disease in each group (primary or recurrent)?

Thank you for the opportunity to clarify. In the results section, line 5,  we reported that “All cases were diagnosed with a primary cancer.”

Regarding the results, in my point of view, the most relevant finding is the significant difference between the rates of wound dehiscence. This finding could be highlighted in the abstract, where is neither cited, and in the discussion and conclusions.

Thank you again for the opportunity to clarify. We agree with you that the significant difference between the rates of wound dehiscence is one of the most relevant findings of our study, allong with the time needed for the complete wound healing and the time to start adjuvant therapy after surgery. As suggested, we highlighted the wound dehiscence rates in the abstract by adding the sentence “inferior perineal wound dehiscence rates (1 versus 9 cases, p=0.033)”  (line 13), in discussion (line 32 – “Precisely, the patients of Group A presented a statistically significant lower incidence of perineal wound dehiscence (1 versus 9 patients, p=0.033”)……” and conclusions (line 5 –“…wound healing associated with inferior dehiscence rates…”).

On the other hand, the difference in the recurrence rate, although twice in the group B, was not significant. I think that the study has no power to identify statistically differences on long term oncological outcomes. So I would just described this findings, but not highlighted as a significant result.

Thank you for the opportunity to clarify.  Regarding the recurrence rates, indeed, the differences between the two groups are not statistically relevant (p=0.152).

In results section (secondary outcomes), we changed the last sentence in: “At a median follow-up period of 30-31 months, 4 patients (16.6 %) in Group A presented a recurrence, while in the Group B that occurred in 12 patients (33.3%) (p=0.152), however, no statistical relevant differences in recurrence and mortality rates were reported in the two groups (table 4).” In discussion section (paragraph 7) we highlight the most relavant oncological outcomes of our study and only describe the recurence and mortality outcomes concluding : “Regarding the long-term outcomes, the recurrence and the mortality rates also resulted lower in Group A (16.6% versus 33.3% and 8.3% versus 11.1%, respectively), but without showing a statistical significance in both cases. Noteworthy, the small number of cases presented limited the statistical power in iden-tifying the differences on long term oncological outcomes.” Finally, in conclusions section we removed the referal to the recurrences and modified as follow : “Furthermore, the quality of life of the patients was improved by anticipating the initiation of the adjuvant treatment and reducing the time needed for perineal wound healing.”